# A Perspective Review on Understanding Drought Stress Tolerance in Wild Banana Genetic Resources of Northeast India

**DOI:** 10.3390/genes14020370

**Published:** 2023-01-31

**Authors:** Surendrakumar Singh Thingnam, Dinamani Singh Lourembam, Punshi Singh Tongbram, Vadthya Lokya, Siddharth Tiwari, Mohd. Kamran Khan, Anamika Pandey, Mehmet Hamurcu, Robert Thangjam

**Affiliations:** 1Department of Biotechnology, School of Life Sciences, Mizoram University, Aizawl 796004, India; 2Plant Tissue Culture and Genetic Engineering Lab, National Agri-Food Biotechnology Institute (NABI), Department of Biotechnology, Ministry of Science and Technology (Government of India), Sector 81, Knowledge City, S.A.S. Nagar, Mohali 140306, India; 3Department of Soil Science and Plant Nutrition, Faculty of Agriculture, Selcuk University, Konya 42079, Turkey; 4Department of Life Sciences, School of Life Sciences, Manipur University, Imphal 795003, India

**Keywords:** drought stress, tolerance, crop wild relatives, wild banana, Musaceae, northeast India

## Abstract

The enormous perennial monocotyledonous herb banana (*Musa* spp.), which includes dessert and cooking varieties, is found in more than 120 countries and is a member of the order Zingiberales and family Musaceae. The production of bananas requires a certain amount of precipitation throughout the year, and its scarcity reduces productivity in rain-fed banana-growing areas due to drought stress. To increase the tolerance of banana crops to drought stress, it is necessary to explore crop wild relatives (CWRs) of banana. Although molecular genetic pathways involved in drought stress tolerance of cultivated banana have been uncovered and understood with the introduction of high-throughput DNA sequencing technology, next-generation sequencing (NGS) techniques, and numerous “omics” tools, unfortunately, such approaches have not been thoroughly implemented to utilize the huge potential of wild genetic resources of banana. In India, the northeastern region has been reported to have the highest diversity and distribution of Musaceae, with more than 30 taxa, 19 of which are unique to the area, accounting for around 81% of all wild species. As a result, the area is regarded as one of the main locations of origin for the Musaceae family. The understanding of the response of the banana genotypes of northeastern India belonging to different genome groups to water deficit stress at the molecular level will be useful for developing and improving drought tolerance in commercial banana cultivars not only in India but also worldwide. Hence, in the present review, we discuss the studies conducted to observe the effect of drought stress on different banana species. Moreover, the article highlights the tools and techniques that have been used or that can be used for exploring and understanding the molecular basis of differentially regulated genes and their networks in different drought stress-tolerant banana genotypes of northeast India, especially wild types, for unraveling their potential novel traits and genes.

## 1. Introduction

Drought is a major threat arising out of unstable environmental conditions and affecting the global agricultural sector and population [1,2]. It is affecting 28% of the soil biosphere, causing hollowness, reduced water holding capacity, and additional factors that threaten growth and development, causing a decrease in crop yield [3,4]. Banana production is also hugely affected by the complex biotic and abiotic stresses around the globe [5], including drought, low soil fertility, heat, and salinity [6,7,8,9], among which drought is one of the most prevailing threats [10]. Banana grown in humid, warm environments of tropical and subtropical regions has a permanent green canopy and shallow roots that are highly sensitive to drought [9,11]. It is reported that annual rainfall below 100 mm is associated with a 65% reduction in banana yields [9,12]. Proper banana production requires evenly distributed rainfall between 1200 and 2600 mm per year [13]. Even though commercial plantation programs meet the water requirement through various irrigation methods, they only contribute to 15% of the global production [14,15]. Drought leads to dehydration, osmotic imbalance, stomatal functions, photosynthesis, and overheating of cells and tissues that affect the normal growth, development, and reproduction of bananas [16,17,18]. However, stress-tolerant plants possess multiple adaptive mechanisms, including at the physiological, biochemical, cellular, and molecular levels, to protect themselves from drought stress. Therefore, an improved banana breeding program utilizing stress-tolerant genotypes is essential for enhancing yields under drought conditions [19,20]. For an upgraded breeding plan, it is necessary to find the banana genotypes that may generate a considerable yield when provided with less water [21]. Crop Wild Relatives (CWRs) are the closely associated species of cultivated genotypes with numerous stress-tolerance traits that have been neglected while focusing on better yields, cost-effectiveness, and edibility [22]. With their several specific traits, these are among the resources with the greatest prospect for resolving the issues with sustainable crop production and global food security. Thus, the CWRs should be properly characterized to understand their true potential by utilizing genomic approaches and other cutting-edge biological technologies [23,24]. Most CWRs of *Musa* spp. are seeded and propagated by both sexual and vegetative methods, while the cultivated bananas are sterile and parthenocarpic [25]. Though these crops have poor agronomic traits, researchers are increasingly focused on this crop due to its naturally acquired resistance and tolerance of biotic and abiotic stresses [26]. CWRs of sunflower [27], sorghum [28], barley [29], wheat [30], and alfalfa [31] have been widely screened for drought tolerance. However, banana CWRs remain unexplored and poorly characterized, both phenotypically and molecularly [32]. There is plenty of diversity in the banana CWRs [33] that needs to be tackled for conservation and characterization [34,35].

The use of technological advancement in high-throughput phenotyping, transcriptomics, and proteomics helps to identify and understand the genetic basis of drought stress tolerance in the genetic resources of CWRs [36,37,38,39,40].

The incorporation of stress-tolerant genes, which are naturally present in banana CWRs, is required for producing improved varieties with better performance [41,42]. Drought tolerance is a complex trait because most plants evolve through a series of modifications influenced by environmental factors over location and time, which are controlled by many genes throughout the evolution process [10,35,43]. For tolerance, plants have to induce deep rooting, conservative water use, symbiotic association, and lifecycle modifications to adapt to the changing climate [15,36]. Conventional breeding for drought tolerance in bananas might require several generations of backcrossing. Technological advancement in recent years has also enabled the introduction of desirable genes into the genomes of different genotypes [35]. The wild relatives of plants are known for their tolerance to various biotic and abiotic stresses that occur through the course of evolution [43]. Therefore, identification and molecular characterization of drought-tolerant traits in CWRs can pave the way for generating drought-tolerant banana cultivars through breeding programs and advanced genome editing technology [44,45].

In 2019–2020, global banana production has been reported to be around 119 million metric tons, with 30 million metric tons produced by India (https://www.fao.org/faostat/en/#data accessed on 22 November 2022). There are over 300 different types of bananas worldwide (Figure 1). However, farming in India primarily uses 15–20 types (Figure 2). In India, Musaceae are the most diverse and widely distributed in the northeastern states of India, where there are 30 taxa, 19 of which are unique to the area. This accounts for around 81% of India’s entire diversity of wild Musaceae [46]. This further supports the idea that this area is one of the primary places of origin for the family Musaceae and shows that the area bordering Bangladesh, China, and Myanmar is a biodiversity-rich area for the Musaceae. According to the distribution modeling approach, the largest area of banana CWR species richness was identified in the northeastern states of India, on the Malay Peninsula, and near the border between south China and northern Vietnam. It was discovered that 9 out of the 59 analyzed species are endangered, and 11 can be classified as vulnerable [47]. In addition, evaluation of in and ex-situ conservation status shows that 49 of the 59 studied species are of high importance for additional conservation, and 56 of them are now inadequately conserved ex-situ [48]. Due to the importance of banana’s diverse genetic resources, in this review we discuss research conducted in different banana species under drought stress, with a specific focus on the banana genotypes of the northeastern region of India, which is considered one of the significant locations of origin of the Musaceae family.

## 2. General Description and Taxonomy of *Musa*

Bananas (*Musa* spp.) are monocotyledonous herbs of the *Musaceae* family of the *Zingiberales* order [49] cultivated in more than 130 countries in the tropical and subtropical regions of Asia, Africa, the Caribbean, Latin America, and the Pacific, making it the fourth largest significant crop after rice, wheat, and maize [12]. Some phanerogamists believe Musaceae is composed of only the two genera *Musa* and *Ensete* in the family, while others have reported that the family has six genera and 130 species [50,51]. The genus *Musa* can be divided into four sub-divisions based on chromosome number: Australimusa, Callimusa, Eumusa, and Rhodochlamys, all of which contain edible cultivars. Therefore, individuals with eleven chromosomes (2n = 22) exhibit the traits of Rhodoclamys and Eumusa. Additionally, the traits of Australimusa and Callimusa can be distinguished while having ten chromosomes (2n = 20). The majority of cultivars are members of the largest and most widespread geological section, known as Eumusa, and are descended from *M. acuminata* Colla and *M. balbisiana* Colla) [52]. These two plants, *M. acuminata* Colla and *M. balbisiana* Colla, which yield fruits, are also known by the common name “banana”. Even though there are many different banana varieties, such as the Petite Naine, Poyo, Williams Lacatan, and Grande Naine, the subgroups of *M. acuminata* known as Cavendish and Plantain are prominent in the global banana market.

Cultivated bananas are developed from the two ancestors *M. acuminata* (A) and *M. balbisiana* (B) by intra- or inter-specific crossing, complex initial cultivation, and dispersal over time, resulting in domesticated parthenocarpic, sterile, polyploid cultivars (AA, AAA, AAB, ABB, BB, AB, BBB, AAAA, and ABBB) [25,53,54]. Edible triploid banana cultivars are sterile, genetically uniform, and parthenocarpic, thus causing a bottleneck for conventional breeding techniques [55,56]. However, the production of improved strains based on superior cultivars is supported by biotechnological advances in genetic modification, which removes reproductive barriers and provides a potential substitute for conventional breeding in bananas [57,58].

In India, the largest area under banana cultivation is in Tamil Nadu, followed by Maharashtra, Gujarat, Andhra Pradesh, Karnataka, and other states following the specific trend of cultivating different banana varieties (Table 1).

## 3. Productivity

The global production of staple, commercial fruit crop bananas is about 113.9 million tons and the major production stack is shared by two countries: India (36.7%) and China (13.8%) (https://www.fao.org/faostat/en/#data accessed on 22 November 2022). The preliminary results of FAO-2021 on the banana market review ascertained that Indian banana exports increased significantly from 2018 to 2020 by 233%, or 2.3-fold (Figure 3A). In the fiscal year 2021, fresh bananas worth 7.4 billion Indian rupees were exported from India, which is a significant increase as compared to the year 2018, which was found to be 3.47 billion Indian rupees (Figure 3B). The preliminary FAO-2021 results revealed that the global import market share by various countries is as follows: European Union (27%), United States of America (22%), China (9%), Russian Federation (8%), Japan (6%), and Latin America and the Caribbean (4%).

## 4. Importance of Crop Wild Relatives (CWR)

Wild-seeded *M. acuminata* (A) and *M. balbisiana* (B) were crossed intra- or inter-specifically to produce domesticated banana cultivars [59]. The banana varieties that constitute the ‘A’ genome produce more yield with longer, larger fingers and are more durable in the green and mature phases because they are native to humid, warm climates [60]. It also indicates that genome ‘A’ contains more genes that are important for high yield production and even have qualities that can be used as candidate genes in breeding programs [61]. In contrast, the ‘B’ genome contains more tolerant genes, as they are known to have originated from dry and high-temperature regions, which possess harsher environmental conditions [62,63]. Several experiments have reported that the banana genome group with the ‘B’ genome has the capability to withstand more abiotic stress, such as wide temperature variations, low soil water content, and salt or environmental stresses [17,64]. Wild-seeded relatives of edible bananas serve as a reservoir for a diverse gene pool of resistant genes against major diseases, pests, or abiotic stress, which can be used for crop improvement and adaptation [62,65,66].

*M acuminata* (A genome) had the highest intra-species diversity and was classified into ten subspecies [67]. *M acuminata* ssp. *burmannicoides* has been used in various crosses for improved traits as it is tolerant to black Sigatoka and Fusarium wilt race 1 [64]. Most of the wild subspecies are reported from Asia, such as *M. acuminata* ssp. *macrocarpa,* ssp. *zebrina*, ssp. *burmannica*, ssp. *malaccensis*, ssp. *burmannicoides*, ssp. *errans*, ssp. *truncata*, ssp. *siamea*, and ssp. *banksii* [59,67,68]. *M. balbisiana,* another important wild progenitor of ‘genome B,’ is also used as a source of stress-tolerant genes, such as Xanthomonas resistance [66] and drought [69]. Several other wild-seeded *Musa* spp. have also been reported, such as *M. haekkinenii* [70], *M. itinerans* [71], and *M. serpentina* [72]. Among them, the *M. serpentina* species found along the border of Myanmar shows the characteristics of both *M. acuminata* and *M. laterita* [72]. The diversification of wild banana species spreads over a large area extending from Queensland, Australia, to southern China [51]. The progenitors of *M. balbisiana* have highly intraspecific diversity, even though there are no variable subgroups reported so far, while ten subgroups have been identified within *M. acuminata* [61]. It also demonstrated that within cultivars there are three to five ancestral subspecies across the chromosomes, creating “mosaic” patterns of subspecies-specific regions. It has been found that there are three main *M. acuminata* subspecies that are the main ancestors of the A genome cultivated bananas: *M. acuminata* ssp. *banksii*, *M. acuminata* ssp. *zebrina*, and *M. acuminata* ssp. *malaccensis* [59]. Several ancestors, however, remain to be identified [73,74]. The use of wild bananas for various purposes, including as a food source, a building material, a tool for handicrafts, and a traditional remedy for ailments such as dysentery, diabetes mellitus, heartburn, diarrhea, and stomach cramps, as well as anemia, malaria, snakebite fevers, burns, inflammation, and pains, has been documented in earlier Indian medical epics and other documents [75,76]. In addition, wild bananas provide a valuable source of genetic diversity that could be used to advance banana cultivation [77].

*M. acuminata* subsp. *burmannica*, *M. balbisiana*, *Musa basjoo*, *M. itinerans*, *M nagensium*, *M. ruiliensis*, *M. velutina*, and *M. yunnanensis* were found to be tolerant against Foc-TR4 [78], whereas spp. *burmannica* is tolerant to Sigatoka leaf spot diseases [79]. Although it lacks a distinct subspecies designation, the wild *M. balbisiana* species is composed of a variety of wild types and is a fantastic source of genes that are resistant to a wide range of biotic or abiotic challenges. It is resistant to Fusarium wilt, Sigatoka, Septoria, and Cordana leaf spots, rust, and bacterial diseases, such as head rot (*Erwinia* spp.), in addition to tolerance for the rhizome weevil (*Cosmopolites sordidus* Germar) and pseudostem weevil (*Odoiporus longicollis*) [80]. Abiotic challenges, such as cold, water deficits, and unfavorable soil conditions, are also highly tolerated by *M. balbisiana*, in addition to biotic stresses [81].

The genotype of the plant has the potential for stress tolerance; however, water management, good scientific cultivation practices, and crop management can minimize the gap between maximal and actual production [82]. Explorations in India’s northeastern regions have revealed the presence of numerous novel species, mutants, and natural hybrids, in addition to adverse terrain, favorable political climates, inadequate transportation infrastructure, and other factors that make it challenging to reach the natural habitats of the *Musa* [83].

## 5. Wild Relatives of *Musa* in India

The family Musaceae is represented by the two genera, *Ensete* and *Musa*, in India, and 37 wild taxa have been reported, with the majority of them occurring naturally in northeast India [47,84]. Figure 4 shows the distribution of the wild Musa taxa in the northeastern states of India. Three subspecies of *M. acuminata*, *burmannicoides, burmannica,* and *banksii*, have been reported in India, occurring in the south and middle Andamans, Western Ghats (Karnataka), Kaziranga Forest (Assam), and Khasi hill ranges (Meghalaya) [83]. These relatives of wild bananas are potentially valuable first sources of genes that could confer resistance to a wide range of pathogens and pests [83]. However, *M. balbisiana* is found in northeastern states, including the Andaman and Nicobar Islands, Bihar, West Bengal, Orissa, Kerala, Karnataka, Tamil Nadu, and Andhra Pradesh [83]. Genes related to abiotic stress tolerance, disease, and pest resistance can be found in abundance in the “B” genome pool [85,86]. Due to its low level of diversity, this species cannot be divided into subspecies; instead, it is written as “types,” which refer to the region or location of the collection. Among the inhabitants of Assam, the Western Ghats of Kerala, and the people of Karnataka, *M. balbisiana* variants have taken on the function of commercial variations [83].

Among the 37 known species of *Musa* (sect.), seven have been reported in India: *M. sikkimensis, M. ingens, M. nagensium, M. cheesmani, M. basjoo*, *M. itinerans*, and *M. flaviflora* [87,88]. *M. itinerans*, a rare species, has been reported from Manipur, Arunachal Pradesh, and Mizoram [89]. *M. nagensium* is frequently found in the damp evergreen forests of the northeastern states of Meghalaya, Arunachal Pradesh, and Manipur. In Arunachal Pradesh, *M. nagensium* [88] is distributed monotonously over hundreds of acres, where *Musa aurantiaca* is also present [89]. *M. nagalandiana*, a very rare species of the genus Musa, is named after Nagaland, where it was first identified, and is found growing on hillside slopes in semi-evergreen tropical forests close to the Doyang river in the Zunheboto region of Nagaland [90]. Rajib Gogoi named the species *M. puspanjalia* in honor of his mother, Puspanjali Gogoi. Large groups of this species are frequently observed on stream banks and hill slopes. Another northeast Indian species, *M. kamengensis*, is frequently observed on the Jamiri mountain slopes at a height of 1500 m in the West Kameng region of Arunachal Pradesh. It appears to be cold-tolerant [91]. The first *M. aurantiaca* specimen was discovered in Assam, and the variety name homenborgohainiana was derived after Homen Borgohain (an Assamese writer) [92]. The variety *M. aurantiaca* var. *jengingensis* was given its name after Jenging, the location where it was originally noticed. Quite different from the others, the male inflorescence of this variety vanishes before the fruits reach maturity [92]. The first specimen to be described was *M. mannii* var. *namdangensis* of Assam, which took its name from Namdang, the location of its origin, and was only observed in this region. [93]. The Indian regions in the North-Eastern States, the Western and Eastern Ghats, and the Andaman and Nicobar Islands are known for the occurrence of high wild Musa species diversity. However, due to the region’s lush evergreen forests, topography, and civil upheaval, systematic exploration of northeastern India has been fairly limited, leaving many more taxa yet to be identified [94,95].

## 6. Strategies for Evaluating Drought Stress Response of Wild *Musa* spp. Found in Northeast India

The occurrence of a huge reserve of diverse wild *Musa* spp. genetic resources naturally in the northeast Indian region, which have remained mostly uncharacterized and untapped, provides vast opportunities for the evaluation of various novel traits, such as stress tolerance during drought. The following is a summary of strategies for the evaluation of drought-induced responses in these different genotypes.

### 6.1. Field Studies

Field studies are used for monitoring the stress-responsive adaptation of the various plants to the conditions applied, and such an approach will be useful in the drought-response studies in the wild *Musa* spp. found in northeast India (Figure 5A). In banana, to obtain consistency in development, tissue culture plants with 4–6 healthy leaves and 4–5 primary cord roots or uniform-size suckers weighing 1.5–2.0 kg should be planted [6]. It is important to take precautions to conduct such field studies in uniformly structured and fertile soil.

### 6.2. Pot Studies

Three-month-old nursery plants or plantlets produced by tissue culture should be transferred and grown in 50 L plastic pots with peat substrate in a greenhouse. K_2_O (300 mg/L), P_2_O_5_ (200 mg/L), and N (200 mg/L) should be added to the substrate as nutrients (Figure 5B). The greenhouse should be maintained under conditions of 20–30 °C, 70–90% RH, 200 μmolm^−2^s^−1^ of light intensity, and a 16 h light/8 h dark cycle [15]; or soil supplied with half-strength Hoagland’s solution [96,97] or coconut coir medium [98] or in the normal soil without any supplements [57].

#### 6.2.1. Induction of Drought Stress

For both the field and pot experiments, uniform sized plants having 5–6 healthy leaves and 4–5 major roots in 70–80 kg capacity pots filled with equal amounts of soil, sand, and compost [10] should be selected for drought stress treatments. Though the induction of drought stress in experiments depends on several factors, such as the size of the soil in the experiment, soil moisture, the growth environment, etc., different studies have developed drought stress by withholding water for 8 to 45 days [15,99,100,101,102] to monitor the soil matric potential and leaf gas exchange parameters under drought stress.

#### 6.2.2. Phenotypic Evaluation of Drought-Stress Response

To choose stress-resistant cultivars and improve stress management techniques, plant stress phenotyping is crucial. For enhancing traits, it is crucial to make an efficient phenotyping plan. The following experimental plans would be helpful to assess the phenotypic observations under drought conditions and could be exploited for further genetic improvement of cultivated varieties of banana. Plants maintained under drought stress conditions should be evaluated for the following physiological responses:Leaf area, number of emerging leaves, stem circumference, and height [103]Relative water content [15,97,104]Relative growth rate [10]Net assimilation rate [105]Leaf water retention capacity [106]Leaf gas exchange [107]

### 6.3. In Vitro Plants for Evaluation

The in vitro regenerated-plantlets with 4–5 leaves should be used for studies that evaluate drought stress responses [108]. Drought conditions can be mimicked in an in vitro environment by treating the plants with various concentrations of PEG for 1, 6, and 24 h [109] or 50 to 100 mM mannitol [99]. Figure 5C shows the steps for the evaluation of the drought stress response under in vitro conditions.

## 7. Physiochemical Parameters for Evaluation of Drought-Stress Response

For the evaluation of the drought-stress responses by the plants under field, pot, and in vitro conditions, the following analysis should be conducted:Quantification of photosynthetic pigments [110]Yield and yield parameters [101]Drought susceptibility index [111]Malondialdehyde (MDA) content [97]Hydrogen peroxide (H_2_O_2_) activity [98]Proline content [112]Soluble sugar content [98]Endogenous Abscisic acid (ABA) content [98]Photosynthetic pigments [110]Electrolyte leakage [113]Photosynthetic efficiency [114]Total glutathione content [115]Leaf disc senescence assay [116]Superoxide dismutase (SOD) activity [117]Glutathione reduction activity [118]Catalase activity [119]

## 8. High-Throughput Phenotyping

In comparison to visual measurement, the reliability and accuracy of stress evaluation have increased with the application of imaging tools and the standardization of visual assessments. However, a large number of wild banana species are still poorly characterized and not evaluated for important new allelic diversity for stress tolerance. In a recent study, dynamic high-throughput phenotyping, modeling, and validation techniques were employed to evaluate genotype-dependent transpiration responses to variable environmental conditions in diverse wild banana genotypes and subspecies containing known ancestors of banana cultivars [38]. The high-throughput phenotyping can also monitor real-time transpiration, measurement of leaf gas exchange, leaf temperature, and use of a leaf patch pressure probe, which is effective in measuring plant water status in banana [120,121]. The diverse banana accessions maintained at CGIAR Genebank International under the Musa Germplasm Transit Centre (ITC) are key sources for phenotypic evaluation of desirable agronomic traits that could be used for cultivar selection and breeding tolerant to biotic and abiotic conditions [37]. With the expanding capabilities of machine learning (ML) techniques, new insights can be extracted from curated, annotated, and high-dimensional datasets spanning a variety of crops and conditions [122]. Furthermore, deep learning (DL), a subdivision of ML methods, is considered an adaptable strategy for high-throughput phenotyping and genotyping, as it helps in the identification and quantification of plant stress [60]. Therefore, high-throughput phenotyping can be effectively employed to identify drought-tolerant traits in widely available CWRs of banana from northeast India.

## 9. Studies Employing Breeding and Omics Approaches

Scientific and information technology advancements are now being used to generate and process large biological data sets derived from omics studies. The various applications of “omics tools,” such as genomics, proteomics, and transcriptomics, have promoted a critical shift in the study of biological sciences. Furthermore, genomics and bioinformatics approaches have greatly enhanced our understanding of the genetic diversity of several banana wild relatives and their links to the desired phenotypic traits. Moreover, to describe the cellular phenotype and establish a connection with the distinct banana genotype, the combination of RNA-Seq, proteomics, and metabolite studies is very helpful [123].

Information produced by the genomics of CWRs facilitates their use to increase the genetic variability of crop plants. The efficient genome sequencing of CWRs and their effective usage in agricultural improvement are made possible by advancements in DNA sequencing technology [124]. With the advancement of DNA sequencing technology and the availability of next-generation sequencing (NGS) techniques, it has been possible to overcome the limitations of conventional DNA sequencing methods.

The Banana Genome Hub (https://banana-genome-hub.southgreen.fr/, accessed on 22 December 2022) provides the modules to facilitate the integration of various systems that include several major banana datasets [125]. Similarly, the availability of the banana reference genome sequence and the Biodiversity International Musa Germplasm Transit Centre (ITC, https://www.biodiversityinternational.org/banana-genebank/, accessed on 22 December 2022) are helping to explore the banana resources for their genetic variability. The ITC is the world’s largest collection of banana germplasm (1617 accessions of edible and wild species of banana from 38 countries) and was established with the support of the FAO to secure the long-term conservation of the entire banana gene pool for the benefit of future generations. It also ensures that germplasm is free from pests and diseases and available under the International Treaty on Plant Genetic Resources for Food and Agriculture. It also acts as a crucial safety backup and transit center for national banana genebanks. The ITC has provided more than 18,000 accession samples of bananas during its more than 35-year existence to researchers and farmers in 113 countries [37]. Research that increases our understanding of the genetic diversity of bananas has been accelerated by Next Generation Sequencing (NGS) technologies [126]. In addition to providing breeders and researchers with a large amount of information essential for further genetic improvement, the advantage of NGS technologies provides a unique opportunity to harness the genetic variability available in genebanks. The genomic analysis of NAC transcription factors in *M. acuminata* was compared with the NAC genes of *Arabidopsis thaliana*, *Vitis vinifera*, and *Oryza sativa* [127]. The orthologous associations identified in their study can be employed as a reference for NAC function studies. Although molecular markers have been used in banana breeding studies for diversity analysis, they mostly remained unexplored in drought stress linked marker-assisted selection [17]. However, their efficient employment is required to link the drought tolerance in bananas with specific markers.

## 10. Transcriptomics Analysis

The current technological advancement in transcriptomic studies has huge potential for the identification of stress-responsive genes that govern a particular trait [128]. The availability of numerous wild, uncharacterized banana genetic resources in northeastern India has the potential to reveal novel genes that are associated with several stress-tolerant characteristics, such as drought stress. Analyzing and comparing the transcriptomic response of these unique wild accessions with the susceptible varieties can generate useful information on drought tolerance at the gene level that can be applied to the development of drought-tolerant bananas. Comparative analysis of transcriptomes that are generated by drought-susceptible and drought-tolerant plants in response to drought stress involves the extraction of the total RNA, preparation of the cDNA library, deep sequencing, analysis and identification of the differential gene expression, and functional validation [128,129,130,131,132]. A study on three triploid banana genotypes grown under control conditions and a mild osmotic stress (5% PEG) by evaluating the transcriptome response to osmotic stress in roots using 18 mRNA-seq libraries revealed that the roots change the spectrum of energy metabolism in response to osmotic stress, leading to a drop in energy level by metabolic shift [132]. mRNA-Seq was used for tracking the transcriptomes of sensitive (Grand Naine, AAA genome) and drought-tolerant (Saba, ABB genome) banana cultivars under controlled and drought stress conditions [129]. They not only discovered 991 upregulated and 1104 downregulated differentially expressed genes (DEGs) in the tolerant cultivar, as well as 1378 upregulated and 1583 downregulated genes in the sensitive cultivar, but they also observed genotype-dependent gene expression patterns in both the cultivars in response to drought stress. The combined transcriptomics and proteomics study also indicated that the roots of the three triploid (ABB) bananas change the broad spectrum of energy metabolism, with 42% and 62% longer roots in the control condition and osmotic stress condition, respectively [133]. Transcriptome analysis of the roots of 12 different triploid banana genotypes representing the AAA, AAB, and ABB genomes revealed that allopolyploidy induces changes in the genome structure that further influence the pattern of gene expression [127]. Overall, the transcriptome analysis with the help of bioinformatics has huge potential for the identification and characterization of specific potential traits in banana CWRs.

## 11. Proteomics Studies

Using an in vitro growth model, the screening of drought-tolerant varieties and their mechanisms in different *Musa* genotypes can be studied through proteomics. Proteome analysis was successful in the investigation of drought-stressed bananas, which revealed a new balance and played an important role in the metabolism of reactive oxygen species (ROS), and several dehydrogenases are involved in NAD/NADH homeostasis and respiration [69]. Thus, the in vitro model can also be used for evaluating the poorly studied wild *Musa* accession of northeast India to understand genotype-specific stress tolerance. Two-dimensional gel electrophoresis has an important application in studying the differential expression of stress-responsive proteins [69,133]. Furthermore, the technological advancement in mass spectrometry has the potential to identify a large number of proteins with high accuracy [134]. Using high-throughput quantitative LC-MS proteomics along with transcriptomics, 2749 root proteins associated with osmotic stress were identified in three banana cultivars [36].

## 12. Genome Editing

The identification and characterization of drought tolerance genes provide a huge opportunity for successful banana genetic improvement through the intervention of biotechnological advancements. To create designer crops with desired features, biotechnological techniques, such as gene editing or genome editing, are extensively used [45,135,136]. For a precise cleavage in the desired target area of the genome to achieve genetic modifications, site-directed nuclease (SDN) approaches have also been used recently [137]. Currently, a few SDN tools are employed for gene editing, including Zinc-Finger Nucleases (ZFNs), Transcription Activator-Like Effector Nucleases (TALENs), and Clustered Regularly Interspaced Short Palindromic Repeats (CRISPR) systems [138,139].

Among such editing technologies, CRISPR/Cas-mediated tools have emerged as the most potent and precise methods for crop genome editing, originally derived from the *Streptococcus pyogenes* adaptive immune system [139]. The CRISPR/Cas9-mediated gene editing technique is composed of guide RNA (synthetic gRNA) and Cas9 nuclease, which performs precise double-strand breaks (DSBs) at 3-4 nucleotides upstream of PAM after the target genomic sequence is recognized by gRNA. Following, a possible natural repair mechanism occurs either by non-homologous end-joining (NHEJ) or homology-directed repair (HDR), which results in small insertions or deletions, substitutions of nucleotides, or gene replacement. In addition to Cas9, other Cas proteins, such as Cas13a and Cas12a, which perform edits in the plant genome used for multiplexing, can also target single-strand RNA [45]. Additionally, a new type of editing system has been developed with increased precision, such as prime editing, which can perform DNA base pair substitutions, small insertions or deletions [140], and base editing, used for even a single base pair edit at the target site without the requirement of DBS and HDR templates [141,142]. The development of genetically altered and genome-edited crop plants involves the development of an extremely effective transformation and regeneration system [43]. The routine techniques used for genetic transformation include Agrobacterium-mediated transformation [43,143,144,145], protoplast transformation [136,146], and particle bombardment [147]. In the case of bananas, the preferred starting material for genetic transformation is embryogenic cells (ECs), and their regeneration systems have been well-developed for several banana species [43,148,149,150,151]. So far, many research groups have developed ECs for successful genetic transformation and regeneration protocols for cultivar-specific banana plants [43,53]. Protoplast transformation is a versatile platform for the generation of DNA-free edited plants with the intervention of a new direct delivery system of the ribonucleoprotein (RNP) complex consisting of gRNA and Cas9 nuclease (152). Nevertheless, the regeneration of plants from banana protoplasts remains a challenge due to in vitro instability, poor reproducibility, and a low regeneration rate [151,152,153].

Despite all the difficulties associated with genetic transformation and regeneration in bananas, several research groups have successfully optimized and executed research studies by targeting some marker genes, such as phytoene desaturase (PDS), to demonstrate the proof of concept of genome editing [145,154]. The PDS gene in the banana cultivar Rasthali (AAB) was precisely and effectively edited as the first experiment of CRISPR/Cas9-mediated genome editing using single guide RNA [145]. Subsequently, the same group of researchers implemented a CRISPR/Cas9-based approach for targeted mutagenesis of *lycopene epsilon-cyclase* to develop the β-carotene-enriched Grand Naine (AAA) banana [43]. Similarly, the PDS gene in the Cavendish Williams (AAA) genome was edited using polycistronic tRNA [154] and the banana cultivar Sukali Ndiizi (AAB) using multiple gRNAs [155]. Furthermore, several genes are identified and targeted through CRISPR/Cas9 gene editing tools for banana genetic improvement, particularly toward disease resistance [139,156,157]. The group of [158] demonstrated the CRISPR/Cas9-mediated genome editing of the aminocyclopropane 1-carboxylate oxidase 1 (MaACO1) gene and extended the shelf-life of banana fruit through the reduced synthesis of ethylene. In a recent study, the CRISPR/Cas9 gene editing tool was used to study the role of the *CCD4* gene in β-carotene accumulation in Rasthali fruit pulp, demonstrating the transgene-free approach to banana genetic improvement [136]. As a result, the identification and functional characterization of drought tolerance traits in the CWR of northeast India aid in the development of elite banana cultivars through genome editing. Following the year 2000, production of genetically modified crops has increased over 100-fold, and globally, farmers are cultivating around 190 million hectares of mostly soybean, maize, cotton, and canola [159,160,161]. The biosafety legislation for risk assessment and management lies at different levels in different countries for commercialization [162,163]. In India, SDN-1 and SDN-2 categories of genome-edited products free from exogenous DNA have been exempt from biosafety assessment (Guidelines for the safety assessment of genome-edited plants, 2022).

## 13. Conclusions and Future Prospects

CWRs are closely related species of domestic genotypes with a high tolerance towards different stress conditions and, thus, need to be thoroughly explored for different traits [164,165]. Bananas are essential food for many developing countries, but unfortunately, one of the main obstacles to banana production is drought. Due to continuously increasing climate change, the negative impact of drought on banana crop production has significantly increased across the tropics and subtropics of the world. Although bananas are highly vulnerable to drought, genotypes with the “ABB” genome are more tolerant of drought stress than other genotypes [10,104]. The enhanced yield of bananas with limited resources of water can be achieved through the development of drought-tolerant banana genotypes using genetic improvement. The development of climate-smart banana varieties through biotechnological strategies depends on the determination of traits conferring genes for drought tolerance in different wild varieties of bananas. Hence, the application of high-throughput phenotyping of wild bananas and adequate NGS data and other omics approaches can pave the way for further genetic improvement for drought tolerance [166]. Subsequently, the intervention of the genome-editing tool CRISPR/Cas can produce drought-tolerant commercial banana cultivars. Since genome-edited plants using Site-Directed Nuclease (SDN) technology are considered part of the SDN1 and SDN2 categories, more relaxed regulations are being adopted for new variety commercialization in India. Therefore, continuous screening or searching for drought-tolerance traits in northeastern Indian wild bananas is vital so that contemporary molecular strategies can be efficiently utilized for the development of climate-smart commercial banana cultivars for sustainable food security. Moreover, the incorporation of biotechnological, conventional, and molecular breeding tools for the utilization of available genetic resources can be a more successful strategy for the development and improvement of stress-tolerant bananas [166]. Studies investigating the drought stress tolerance trait in different banana species should be more focused on the nucleotide level. In addition, the full genome sequence of different Musa species should be determined to effectively utilize these species for the development of drought tolerance in modern cultivars.

## Figures and Tables

**Figure 1 genes-14-00370-f001:**
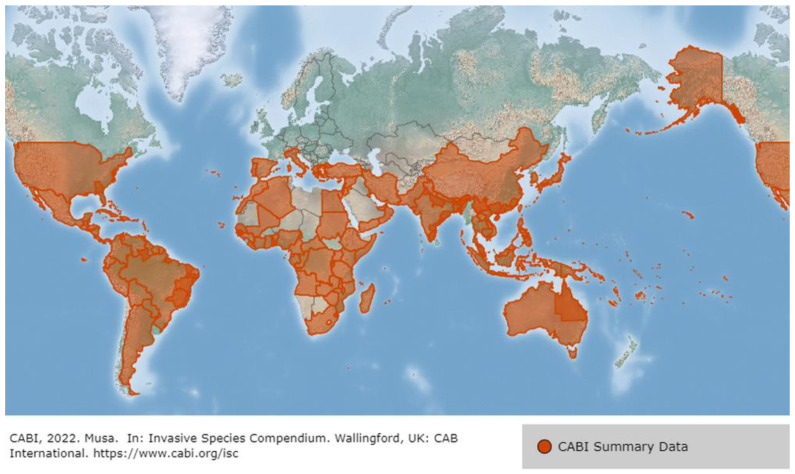
Global distribution of banana (*Musa* spp.) species is indicated by the orange color intensity. Map generated using Centre for Agriculture and Bioscience International (CABI), (https://www.cabi.org/isc/datasheet/35124#toDistributionMaps accessed on 22 December 2022).

**Figure 2 genes-14-00370-f002:**
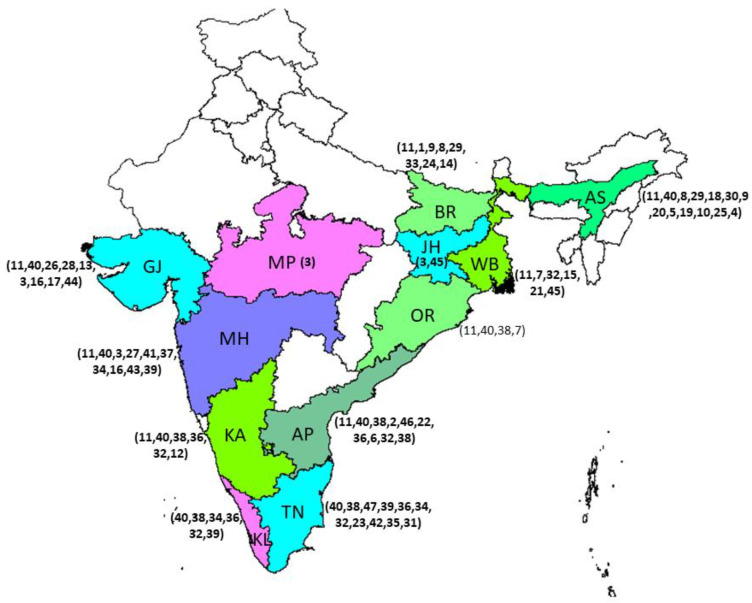
Distribution map of cultivated banana varieties in Indian states generated using DIVA-GIS (http://www.diva-gis.org/gdata accessed on 22nd November 2022). Adopted and modified from the Database of National Horticulture Board, Ministry of Agriculture, Govt. of India (http://nhb.gov.in/report_files/banana/BANANA.htm accessed on 22 December 2022). 1-Alpon, 2-Amritpant, 3-Basrai, 4-Bharat Moni, 5-Bhimkol, 6-Chakrakeli, 7-Champa, 8-ChiniChampa, 9-Chinia, 10-Digjowa, 11-Dwarf Cavendish, 12-Elakkibale, 13-Gandevi Selection, 14-Gauria, 15-Giant Governor, 16-Grand naine, 17-Harichal, 18-Honda, 19-Jatikol, 20-Kanchkol, 21-Kanthali, 22-Karpoora, 23-Karpuravalli, 24-Kothia, 25-Kulpait, 26-Lacatan, 27-Lal Velchi, 28-Lokhandi, 29-Malbhog, 30-Manjahaji, 31-Matti, 32-Monthan, 33-Muthia, 34-Nendran, 35-Peyan, 36-Poovan, 37-Rajeli, 38-Rasthali, 39-Red Banana, 40-Robusta, 41-Safed Velchi, 42-Sakkai, 43-Shreemanti, 44-Shrimati, 45-Singapuri, 46-Thellachakrakeli, 47-Virupakshi, and 48-YenaguBontha.

**Figure 3 genes-14-00370-f003:**
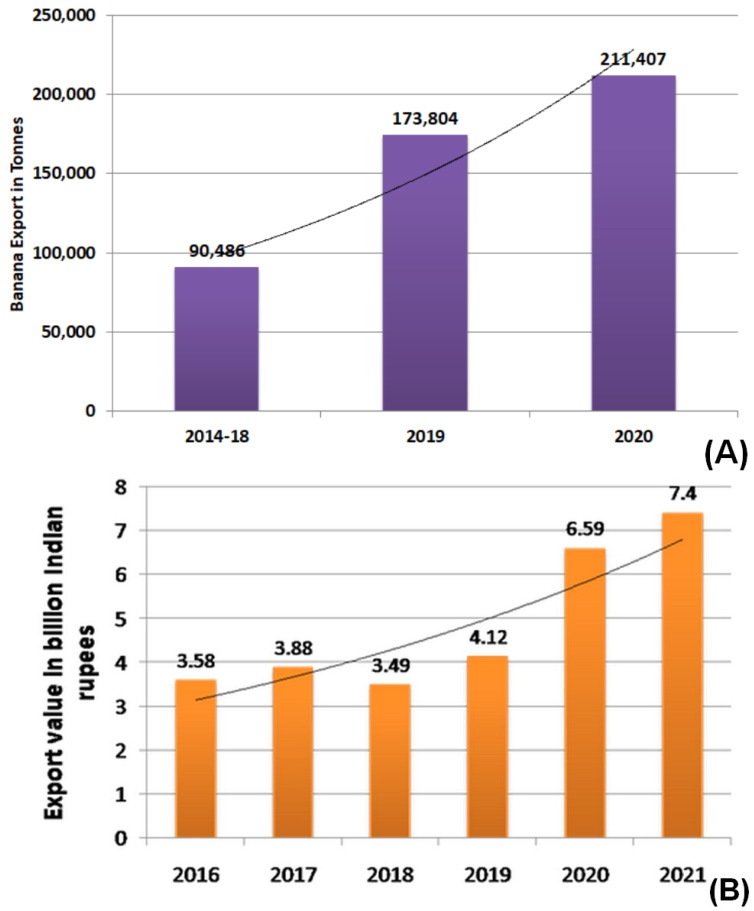
(**A**) FAO-2021 preliminary results on the banana market review for 2020. (**B**) Export value of fresh bananas from India, Fiscal Years 2016–2021 (Published by Statista Research Department, 17 March 2022).

**Figure 4 genes-14-00370-f004:**
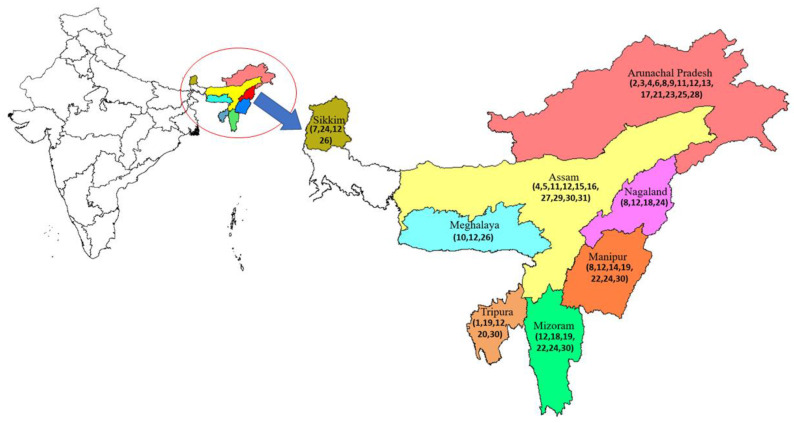
Distribution of wild bananas in northeast India, generated using DIVA-GIS (http://www.diva-gis.org/gdata accessed on 22 November 2022) and adapted from Sabu et al., 2016 [46]. 1-*Ensete glaucum*, 2-*Musa argentii*, 3-*Musa arunachalensis*, 4-*Musa aurantiaca*, 5-*Musa aurantiaca* var. *homenborgohainiana*, 6-*Musa aurantiaca* var. *jengingensis*, 7-*Musa balbisiana* var. *balbisiana*, 8-*Musa cheesmanii*, 9-*Musa chunii*, 10-*Musa cylindrica*, 11-*Musa flaviflora*, 12-*Musa itinerans*, 13-*Musa kamengensis*, 14-*Musa laterita*, 15-*Musa mannii*, 16-*Musa mannii* var. *namdangensis*, 17-*Musa markkui*, 18-*Musa nagensium*, 19-*Musa ochracea*, 20-*Musa ornata*, 21-*Musa pushpanjaliae*, 22-*Musa rubra*, 23-*Musa sanguinea*, 24-*Musa sikkimensis*, 25-*Musa swarnaphalya*, 26-*Musa thomsonii*, 27-*Musa velutina*, 28-*Musa velutina* subsp. *markkuana*, 29-*Musa velutina* var. *variegata*, 30-*Musa balbisiana*, and 31-*Musa balbisiana* var. sepa-athiya.

**Figure 5 genes-14-00370-f005:**
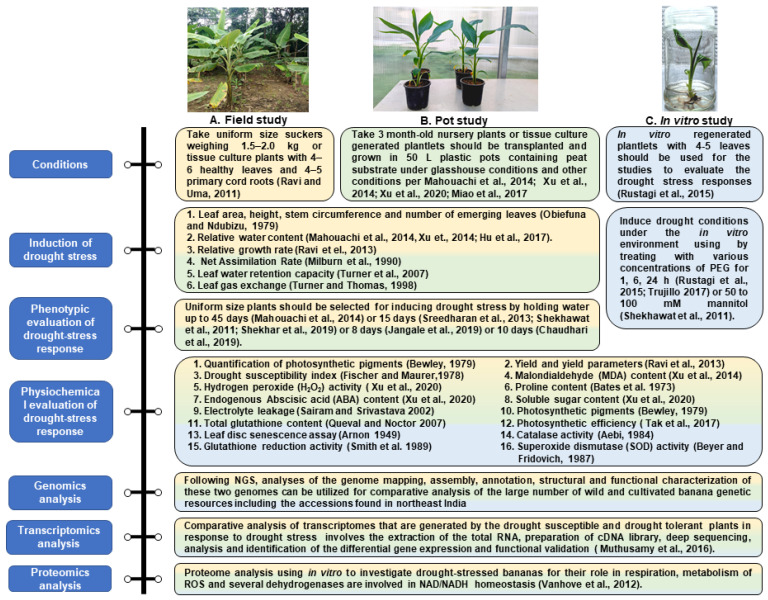
The schematic diagram for evaluating (**A**) Field study, (**B**) Pot study, and (**C**) In vitro study of drought stress response in wild banana genetic resources in northeast India. The figure was adapted from Ravi et al., 2011 [6]; Ravi et al., 2013 [10]; and Vanhove et al., 2012 [69].

**Table 1 genes-14-00370-t001:** Distribution of banana cultivar varieties in Indian states. Adopted and modified from the Database of National Horticulture Board, Ministry of Agriculture, Govt. of India (http://nhb.gov.in/report_files/banana/BANANA.htm accessed on 22 December 2022).

Sl. No.	Indian State	Varieties Cultivation
1	Andhra Pradesh	Dwarf Cavendish, Robusta, Rasthali, Amritpant, Thellachakrakeli, KarpooraPoovan, Chakrakeli, Monthan, YenaguBontha
2	Assam	Jahaji (Dwarf Cavendish), ChiniChampa, Malbhog, Borjahaji (Robusta), Honda, Manjahaji, Chinia (Manohar), Kanchkol, Bhimkol, Jatikol, Digjowa, Kulpait, Bharat Moni
3	Bihar	Dwarf Cavendish, Alpon, Chinia, ChiniChampa, Malbhig, Muthia, Kothia, Gauria
4	Gujarat	Dwarf Cavendish, Lacatan, Harichal (Lokhandi), Gandevi Selection, Basrai, Robusta, G-9, Harichal, Shrimati
5	Jharkhand	Basrai, Singapuri
6	Karnataka	Dwarf Cavendish, Robusta, Rasthali, Poovan, Monthan, Elakkibale
7	Kerala	Nendran (Plantain), Palayankodan (Poovan), Rasthali, Monthan, Red Banana, Robusta
8	Madhya Pradesh	Basrai
9	Maharashtra	Dwarf Cavendish, Basrai, Robusta, Lal Velchi, Safed Velchi, RajeliNendran, Grand Naine, Shreemanti, Red Banana
10	Odissa	Dwarf Cavendish, Robusta, Champa, Patkapura (Rasthali)
11	Tamil Nadu	Virupakshi, Robusta, Rad Banana, Poovan, Rasthali, Nendran, Monthan, Karpuravalli, Sakkai, Peyan, Matti
12	West Bengal	Champa, Mortman, Dwarf Cavendish, Giant Governor, Kanthali, Singapuri

## Data Availability

Not applicable.

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
