# Peer review of "A Perspective Review on Understanding Drought Stress Tolerance in Wild Banana Genetic Resources of Northeast India"

_genes, 2023, doi:10.3390/genes14020370_

Round 1
Reviewer 1 Report
The manuscript entitled ‘A Perspective Review on Understanding Drought Stress Tolerance in Wild Banana Genetic Resources of Northeast India’ describes the potential of tapping into the wild genetic resources in banana utilizing the advancements in phenotyping and next generation sequencing technologies to unravel the genetic basis of drought tolerance traits and development of climate-smart varieties. The article details the global distribution of banana cultivation, taxonomy and information on wild relative germplasm in India, approaches and parameters to evaluate drought stress response, high throughput phenotyping strategies, availability of molecular resources, and the potential of the state-of-the-art gene editing tools to develop drought tolerant banana varieties for production worldwide. The manuscript effectively captures the latest developments in banana genetic and genomic resources that can be leveraged to explore novel sources of drought tolerant genetics from wild banana germplasm and deserves publication.
However major revisions are required in the aspects below.
1. The manuscript needs to be thoroughly checked and revised for English language and grammar, find some examples below.
Line 69: ‘Despite of their poor’ needs correction
Line 373,374: ‘Further, deep learning (DL), a sub division of ML methods has transpire as a adaptable’
Line 364,365: ‘The high throughput phenotyping can also be monitored real-time transpiration,’
Line 334: ‘the in vitro environment using by treating with’
Line 365: ‘can also be monitored real-time transpiration,’
Lines 387-389
2. Reference section need to be checked for errors and formatting. (eg: Reference 46, line 667)
Author Response
Line no.(s) Details of changes/edits made
10 addition
24-25 improvement made
25-26 changes made as suggested by Reviewer 2
30-35 improvement made
40, 42, 45, 48, 51, 57, 58 improvement made
66,67 improvement made
71,72 changes made as suggested by Reviewer 1
81 – 115 improvement made
133, 147, 170, 171, 173 improvement made
186, 188 – 191 improvement made
207, 214 improvement made
240 changes made as suggested by Reviewer 2
249, 252, 264-282 corrections and improvement made
292-295, 304 improvement made
333, 340, 348 improvement made
364-380 changes made as suggested by Reviewer 1
389,391,406 improvement made
413,414,430, 435, 438, 450 improvement made
463,475,476,479, 484-509 improvement made
540-542, 547 improvement made
554 additional reference
557, 561 improvement made
564 additional reference
565-567, 570,571 improvement made
679-682 improvement made
962-969 additional references

Reviewer 2 Report
line 24 and 99: replace with " 81%"
quality of figure 3 very low
line 233: remove typo "Musa"
Line 239: south Remove typo S is capital
Line 245: remove typo "Abiotic stresses"
line 266: remove typo "Musa" go through thr whole manuscript.
line 328: remove typo, "Net assimilation rate"
line 344" H2O2, subscript
please check allabberivation.
Author Response

(The authors gave the same response as above.)

Round 2
Reviewer 1 Report
The authors have made corrections to the manuscript as per recommendations.
Author Response
The following changes are being made as indicated below
Figure 4 and 5: The relevant references from which the figures were adopted and constructed has been appropriately cited in the captions
Line no. changes made
65-74 The sentences have been revised with improved English and grammar
110 nine changed to 9
145-152 Musa abbreviated as M. as suggested
177-180 The sentences have been revised with improved English and grammar
213 The sentences have been revised with improved English and grammar
248 “in” is deleted
268-271 The sentences have been revised with improved English and grammar
289-91 The sentences have been revised with improved English and grammar
298 “remain” changed to “remained”
303-304 The sentences have been revised with improved English and grammar
312-313 References of the adaptations for figure no. 5 mentioned
372-373, 387 The sentences have been revised with improved English and grammar
395-400 The sentences have been revised with improved English and grammar
402-403 The sentence has been removed as improvement
404-409 The sentences have been revised with improved English and grammar
422-423 The sentences have been revised with improved English and grammar
458, 469, 472 The sentences have been revised with improved English and grammar
475-476 The sentences have been revised with improved English and grammar
487, 496 The sentences have been revised with improved English and grammar
501-503 The sentences have been revised with improved English and grammar
508, 512-514 The sentences have been revised with improved English and grammar
515-517, 525 The sentences have been revised with improved English and grammar
552, 566-568 The sentences have been revised with improved English and grammar
574-575 The sentences have been revised with improved English and grammar
